# The differential impacts of COVID-19 mortality on mental health by residential geographic regions: The Los Angeles Pandemic Surveillance Cohort Study

Chun Nok Lam[ID][1]*, Ryan Lee[1], Daniel Soto[1], Alex Ho[ID][2], William Nicholas[2], Jennifer B. Unger[ID][1], Neeraj Sood[1,3,4]

1 Keck School of Medicine, University of Southern California, Los Angeles, CA, United States of America, 2 Los Angeles County Department of Public Health, Los Angeles, CA, United States of America, 3 Schaeffer Center for Health Policy & Economics, University of Southern California, Los Angeles, CA, United States of America, 4 Sol Price School of Public Policy, University of Southern California, Los Angeles, CA, United States of America

* chunnok.lam@med.usc.edu

**Data Availability Statement:** Data from this study comes from three sources. First, data from the

## Abstract

### Objective

This study examines the association between changes in mental health before and during the COVID-19 pandemic and COVID-19 mortality across geographic areas and by race/ethnicity.

### Methods

A cross-sectional survey was conducted in Los Angeles County between April and May 2021. The study used the Patient Health Questionnaire-2 to assess major depression risk. Participants' home ZIP codes were classified into low, middle, and high COVID-19 mortality impacted areas (CMIA).

### Results

While there were existing mental health disparities due to differences in demographics and social determinants of health across CMIA in 2018, the pandemic exacerbated the disparities, especially for residents living in high CMIA. Non-White residents in high CMIA reported the largest deterioration in mental health. Differences in mental health by CMIA persisted after controlling for resident characteristics.

### Conclusions

Living in an area with higher COVID-19 mortality rates may have been associated with worse mental health, with Non-White residents reporting worse mental health outcomes in the high mortality area.

2021 survey was collected from participants recruited through a propriety database, which is a third-party repository that belongs to the LRW, a Material Company. The authors contracted with the LRW for data services and the data used in this section is third party data. Please contact Jennifer Holland at jholland@isacorp.com for contract details to access the data. Second, data from the 2018 survey is from the LA County Health Survey and is publicly available. Please contact the Health Assessment Unit at the LA County Department of Public Health (LACDPH) at LAHealthData@ph. lacounty.gov to request the same dataset. Third, the COVID-19 mortality data was a customized request provided by the epidemiology unit at LACDPH. Public data are available at the LA County COVID-19 Data dashboard. Please contact Dr. William Nicholas at CHIE@ph.lacounty.gov for customized data request. Study authors cannot legally distribute study data without the authorization from these parties.

**Funding:** Peter G. Peterson Foundation, Prof. Neeraj Sood, Chun Nok Lam Conrad N. Hilton Foundation, Prof. Neeraj Sood Office of the President University of Southern California, Prof. Neeraj Sood Los Angeles County Department of Public Health, Prof. Neeraj Sood, Jennifer B. Unger Centers for Disease Control and Prevention, Prof. Neeraj Sood Keck School of Medicine of USC, Prof. Neeraj Sood W. M. Keck Foundation, Prof. Neeraj Sood The funders had no role in study design, data collection and analysis, decision to publish, or preparation of the manuscript.

**Competing interests:** The authors have declared that no competing interests exist.

## Policy implications

It is crucial to advocate for greater mental health resources in high COVID-19 mortality areas especially for racial/ethnic minorities.

## Introduction

SARS-CoV-2 has caused over 770 million confirmed COVID-19 cases and 6.9 million deaths worldwide [1]. Beyond deaths and hospitalizations, the pandemic has affected all facets of society. Financial uncertainty, school and workplace closures, social distancing, quarantine and isolation, and difficulty in accessing health and social services are just a few of the ways the COVID-19 pandemic has impacted communities [2]. These unprecedented challenges could have severe mental health consequences including increased anxiety and depression [3–5]. Communities of color, low-income families, and other socially disadvantaged and vulnerable populations have been disproportionally affected by COVID-19, resulting in widening health disparities [6–9]. Understanding the mental health impacts on geographic regions hardest hit by COVID-19 mortality is important for long-term recovery efforts and preparation for future pandemics.

Variation in demographic characteristics and social determinants of health across geographic regions is associated with inequities in health burdens of residents [10, 11]. The pandemic exacerbated these underlying social and structural differences, including financial insecurity and economic hardship, social isolation and movement restrictions, trauma from the loss of loved ones, etc. [12–14]. Data from across countries, the U.S. nationwide, and Los Angeles County showed a positive correlation between poverty level and COVID-19 mortality rates [15–17]. Previous studies also found disproportionate increases in mental health symptoms during the COVID-19 pandemic among socially and economically marginalized groups, including low-income and unhoused populations [18, 19], racial and ethnic minorities [20, 21], and immigrants [22]. A better understanding of the role of mental health disparities in overall pandemic disparities could inform interventions to improve health equity across communities.

Racial and ethnic minorities, including Black, Hispanic, and American Indian persons, had substantially higher risk of COVID-19 infection, hospitalization, and death compared to non-Hispanic White persons [23]. Living in communities with high COVID-19 mortality rates can increase community members' exposure to the suffering of others, leading to mental health challenges throughout the community. While these communities already experience greater mental health burden due to existing social and structural barriers [24, 25], findings from a national study show that Black and Hispanic persons, in particular those with lower income, demonstrated greater resilience and maintained better mental health wellbeing than their White counterparts in response to the pandemic [26]. It is important to assess whether this observation varies across geographic areas according to the community burden of COVID-19 mortality.

This study examines the association between COVID-19 mortality and the mental health status of residents living in subregions in Los Angeles County. We hypothesize that while there were existing mental health disparities due to differences in demographics and social determinants of health across regions, the pandemic further increased the disparities, especially for residents living in areas with higher COVID-19 mortality rates. We began by examining mortality rates by race/ethnicity before and during the pandemic to confirm the varying

degree of impact in each geographic region. We further examined whether the impact of the pandemic on mental health status by region differed between White and Non-White residents. We hypothesize that racial and ethnic minorities living in high COVID-19 mortality areas will have larger deterioration in mental health compared to non-Hispanic White residents living the same region. Study findings will add to the evidence on mental health disparities by geography and race/ethnicity to help address long-term mental health effects of the COVID-19 pandemic on these communities.

## Methods

### Study design and sample population

Participants were part of the Los Angeles Pandemic Surveillance Cohort Study [27]. The study sample was recruited between March 22 to April 23, 2021, from a proprietary database by LRW, a Material Company, representative of the Los Angeles County residents with enrollment quotas to approximate the age, gender, income, and racial/ethnic distribution of Los Angeles County. See Sood et al. for details on the sampling design [28]. 1,335 participants were invited to participate in a survey and 1,222 (91.5%) completed the survey. The survey was administered in English and Spanish. All study participants provided written informed consent. The Los Angeles County Department of Public Health Institutional Review Board approved all study procedures (IRB Project No. 2021-01-919). All study procedures have conformed to the principles embodied in the Declaration of Helsinki.

### Study outcome

The study examined participants' mental health status using the Patient Health Questionnaire-2 (PHQ-2), asking participants about the frequency of depressed mood and anhedonia over the past two weeks [29]. Response choices for both items were never, several days, more than half the days, or nearly every day, coded from 0 to 3. A summary score was calculated using the two items, ranging from 0 to 6 points total. We dichotomized the sum score, with a cutoff of 3 or greater to indicate the individual screened positive, a proxy for being at risk for major depression.

### Participant characteristics

The geographic region where participants lived was the primary predictor of interest. We used self-reported home ZIP code within Los Angeles County to identify the Service Planning Area (SPA) of residence [30]. Then, using publicly available SPA-level data on COVID-19 mortality rates from Los Angeles County Department of Public Health (LACDPH) between March 1, 2020, and April 15, 2021 [31], we divided the county into three multi-SPA COVID-19 Mortality Impact Areas (CMIA). During this period, the low CMIA comprised SPAs 5 and 8 which had cumulative age-adjusted COVID-19 mortality rates ranging from 85 to 163 per 100,000 residents. The middle CMIA comprised SPAs 1, 2, and 3 which had mortality rates ranging from 195 to 221 per 100,000 residents. The high CMIA comprised SPAs 4, 6 and 7 which had mortality rates ranging from 266 to 343 per 100,000 residents. Other covariates evaluated included self-reported age, gender, race/ethnicity, annual household income, COVID-19 impact on employment, healthcare coverage, regular healthcare access, ever-tested or tested positive for COVID-19 (self and household members), and vaccination against COVID-19. We dichotomized race/ethnicity as Non-Hispanic White (White) vs other categories (Non-White), which included Asian, Black, Hispanic, American Indian and Alaska Native, and Native Hawaiian and Pacific Islander to achieve greater statistical power for comparisons.

### Los Angeles County Health Survey and mortality surveillance data

We incorporated data from the 2018 Los Angeles County Health Survey (LACHS) to provide a pre-COVID-19 estimate of the Los Angeles County population at risk for major depression [32], using the same PHQ-2 scale described above. LACDPH also provided age-adjusted all-cause mortality rates for 2018 (i.e., pre-COVID-19), and non-COVID-19 and COVID-19-specific mortality rates for March 1, 2020, through April 15, 2021 (i.e., the beginning of the pandemic through the time our survey data were collected), stratified by the three CMIAs. Therefore, we had PHQ-2 data for 2018 and 2021 and mortality data for 2018 and 2020–2021, all by CMIA.

### Statistical analysis

We calculated means and standard deviations for PHQ-2 score by CMIA. We used Chi-Square, ANOVA and Poisson regression for bivariate analyses. We conducted a multivariable Poisson regression model to examine whether average PHQ-2 sum scores were statistically different by CMIA, controlling for age, gender, White vs Non-White residents, and household income. We calculated weighted estimates of the proportions and standard errors of the dichotomized PHQ-2 scores from our study survey, using Los Angeles County 2019 census estimates [33], and compared them to the 2018 LACHS PHQ-2 estimates, by CMIA and White vs non-White residents. To account for the different numbers of months in the two comparison periods, we used the average age-adjusted monthly mortality rate for the months in each time period. All statistical tests were performed in Stata 15 with α set at 0.05.

## Results

### Age-adjusted mortality rates in the pre- and COVID-19 periods, by CMIA and race/ethnicity

Data from LACDPH indicated that there were pre-existing differences in mortality rates between the high and low CMIAs in 2018. The differences were further exacerbated in 2020–2021 largely due to higher COVID-19 mortality in the high CMIA. For example, Table 1 shows that the all-cause mortality rate was already highest in 2018 in what would become the high CMIA during pandemic (50.6 per 100,000 residents per average month in the high CMIA versus 47.1 per 100,000 residents per average month in the low CMIA). In 2020–2021, the difference between the high versus low CMIA increased, largely due to greater COVID-19 mortality in the high versus low CMIA (19.4 vs 9.3 per 100,000 per average month).

However, the patterns were different when the data were disaggregated by White versus Non-White residents with several notable findings. First, the age-adjusted all-cause mortality

**Table 1. Age-adjusted monthly mortality rates\* by CMIA, white vs non-white residents\*\*.**

| | 2018 (Pre-COVID-19 Period) | | | 3/1/2020–4/15/2021 (COVID-19 Period) | | | | | |
| --- | --- | --- | --- | --- | --- | --- | --- | --- | --- |
| | All-Cause Mortality | | | Non-COVID-19 Mortality | | | COVID-19 Mortality | | |
| | Overall | White | Non-White | Overall | White | Non-White | Overall | White | Non-White |
| High CMIA | 50.6 | 58.2 | 48.3 | 54.3 | 57.5 | 53.2 | 19.4 | 8.9 | 21.9 |
| Middle CMIA | 47.8 | 53.5 | 42.0 | 51.2 | 55.2 | 47.0 | 13.1 | 8.3 | 17.0 |
| Low CMIA | 47.1 | 47.3 | 46.0 | 50.4 | 49.1 | 50.7 | 9.3 | 4.7 | 13.7 |

CMIA: COVID-19 Mortality Impacted Areas

\*Number of deaths per 100,000 residents per month

\*\*White: non-Hispanic White; Non-White: Hispanic, Black, Asian, Native Hawaiian and Pacific Islander and American Indian and Alaska Native

in 2018 was greater among White residents compared to Non-White residents across all CMIAs. Second, in contrast to the data from 2018, COVID-19 mortality was greater among Non-White residents compared to White residents across all CMIAs. Third, the disparities in COVID-19 mortality by CMIA was larger for Non-White residents as compared to White residents. These patterns suggest that the pandemic reversed the mortality advantage of Non-White residents especially in the high CMIA; the all-cause mortality rate for Non-White residents in the high CMIA was lower than that of White residents pre-pandemic (48.3 versus 58.2 per 100,000 residents per average month) but was higher than that of White residents in 2021 (75.1 versus 66.4 per 100,000 residents per average month).

## Mental health and demographic characteristics of the study sample, by CMIA

Among the survey sample of 1,222 adult residents of Los Angeles County in 2021, 35% lived in the high CMIA, 43% in the middle CMIA, and 22% in the low CMIA. Table 2 shows that mental health outcomes were statistically different by CMIA. Average PHQ-2 summary score indicated that those living in the high CMIA (1.4 ± 1.8) were more likely to be at risk for depression than those in the middle CMIA (1.1 ± 1.5) and low CMIA (1.1 ± 1.4) (p<0.01).

Other participant characteristics also differed by CMIA (Table 2). Participants who lived in the high CMIA were more likely to be younger (71% 18–49 years of age, vs middle: 62%, low: 58%, p<0.01) and Non-White residents (70%, vs middle: 54%, low: 57%, p<0.01). Those who lived in the high CMIA were also more likely to have ever tested positive for COVID-19 (15%, vs middle: 8%, low: 9%, p<0.01), and were more likely to have any adult or child in their household who ever tested positive for COVID-19 (p = 0.01 and p = 0.03, respectively).

Table 3 shows the results of the multivariable regression model, testing whether mental health differences remained significantly different by geographic region while accounting for differences in sociodemographic characteristics. Participants who lived in the high CMIA continued to demonstrate a significant higher PHQ-2 summary score compared to those who lived in the low CMIA (IRR: 1.2, 95% CI: 1.0, 1.3, p < .05).

## Mental health comparison pre- vs COVID-19 period

Fig 1A and 1B show the proportion of adults at risk for major depression (PHQ-2) by CMIA, comparing data from the 2018 LACHS to our 2021 survey data. There were pre-existing disparities in mental health by CMIA in 2018 (Fig 1A), with a higher proportion of adults at risk in the high CMIA (15.4% [95% CI: 13.5, 17.2]) compared to low CMIA 10.6% [95% CI: 8.4, 12.7]). Mental health worsened during the pandemic. Specifically, the proportion of adults at risk increased significantly from 15.4% (95% CI: 13.5, 17.2) in 2018 to 21.3% (95% CI: 17.3, 26.0) in 2021 in high CMIA (Fig 1A and 1B).

The patterns were somewhat different when the data were disaggregated by White vs. Non-White residents. For Non-White residents, the patterns were consistent with the overall population in that the proportion of adults at risk for major depression in 2018 was higher in the high CMIA (low: 11.5% [95% CI: 8.3, 14.7] vs high: 16.1% [95% CI: 14.0, 18.2], and the difference increased from 2018 to 2021, with the largest impact in the high CMIA (2018: 16.1% [95% CI: 14.0, 18.2] vs 2021: 24.3% [95% CI: 19.4, 30.0]) (Fig 1C and 1D). However, for White residents there was no significant increase in the proportion of adults at risk for major depression, although at low CMIA it was slightly higher in 2021 but with overlapping confidence intervals with 2018. Nevertheless, in 2021 the proportion of adults at risk for major depression was significantly higher among non-White residents compared to White residents in the high CMIA (24.3% [95% CI: 19.4, 30.0] vs 11.6% [95% CI: 6.9, 18.8]).

**Table 2. Mental health outcomes and sociodemographic characteristics of study participants (N = 1222).**

| | | Residential Geographic Regions* | | | |
|---|---|---|---|---|---|
| | **Overall (N = 1222)** | **Low CMIA (n = 272, 22%)** | **Middle CMIA (n = 527, 43%)** | **High CMIA (n = 423, 35%)** | **p-value** |
| **PHQ-2 Major Depression**** | | | | | <0.001 |
| (mean, SD), score range 0–6 | 1.2 ± 1.6 | 1.1 ± 1.4 | 1.1 ± 1.5 | 1.4 ± 1.8 | |
| **Age** | | | | | <0.001 |
| 18–29 | 14% | 10% | 14% | 15% | |
| 30–49 | 51% | 48% | 48% | 56% | |
| 50–64 | 27% | 31% | 27% | 23% | |
| ≥65 | 9% | 11% | 12% | 5% | |
| **Gender** | | | | | 0.075 |
| Male | 39% | 36% | 39% | 41% | |
| Female | 60% | 63% | 60% | 58% | |
| Non-binary | 1% | 1% | 1% | 1% | |
| **Race/Ethnicity*** | | | | | <0.001 |
| White | 40% | 43% | 46% | 30% | |
| Non-White | 60% | 57% | 54% | 70% | |
| **Annual Household Income** | | | | | 0.079 |
| Under $50,000 | 30% | 25% | 29% | 33% | |
| $50,000 to $99,999 | 31% | 29% | 31% | 33% | |
| $100,000 or more | 34% | 41% | 34% | 30% | |
| Prefer not to answer | 5% | 5% | 6% | 5% | |
| **Change of Employment since COVID-19** | | | | | 0.299 |
| Not employed (i.e. retired / student) | 13% | 15% | 13% | 11% | |
| Employed, no change | 47% | 50% | 47% | 46% | |
| Experienced reduced wages | 25% | 24% | 26% | 26% | |
| Experienced unemployment | 15% | 11% | 14% | 17% | |
| **Healthcare Coverage** | | | | | 0.153 |
| Private | 65% | 68% | 63% | 65% | |
| Medi-CAL | 18% | 13% | 18% | 21% | |
| Medicare | 9% | 10% | 11% | 6% | |
| Other government programs | 2% | 2% | 2% | 1% | |
| No insurance | 6% | 7% | 6% | 6% | |
| **Regular Health Care Access** | 81% | 82% | 82% | 78% | 0.128 |
| **Tested for COVID-19, Ever** | 82% | 81% | 81% | 84% | 0.414 |
| **Tested Positive for COVID-19, Ever** | 11% | 9% | 8% | 15% | 0.004 |
| **Vaccinated against COVID, ≥1 Dose** | 79% | 79% | 80% | 77% | 0.495 |
| **Household with Adults Tested Positive for COVID-19, Ever** | 15% | 13% | 13% | 19% | 0.012 |
| **Household with Children Tested Positive for COVID-19, Ever** | 3% | 2% | 2% | 4% | 0.030 |

*Low CMIA (SPA 5, 8), Middle CMIA (SPA 1, 2, 3), High CMIA (SPA, 4, 6, 7)

**0-not at all worried, 1-slightly worried, 2-moderately worried, 3-very worried, 4-extremely worries

***White: non-Hispanic White; Non-White: Hispanic, Black, Asian, Native Hawaiian and Pacific Islander and American Indian and Alaska Native

CMIA: COVID-19 Mortality Impacted Areas

**Table 3. Multivariable Poisson regression analysis of PHQ-2 score\* (N = 1222).**

| | IRR | 95% CI | p-value |
|---|---|---|---|
| **Residential Geographic Region** | | | |
| Low CMIA | Ref | Ref | |
| Middle CMIA | 0.99 | 0.86, 1.14 | 0.891 |
| High CMIA | 1.15 | 1.00, 1.32 | 0.045 |
| **Age** | | | |
| 18–29 | Ref | Ref | |
| 30–49 | 0.85 | 0.74, 0.97 | 0.020 |
| 50–64 | 0.72 | 0.61, 0.85 | <0.001 |
| ≥65 | 0.57 | 0.45, 0.72 | <0.001 |
| **Gender** | | | |
| Male | Ref | Ref | |
| Female | 1.11 | 0.99, 1.23 | 0.062 |
| Non-binary | 1.44 | 0.90, 2.30 | 0.133 |
| **Race/Ethnicity\*\*** | | | |
| White | Ref | Ref | |
| Non-White | 1.11 | 0.99, 1.24 | 0.081 |
| **Annual Household Income** | | | |
| Under $50,000 | Ref | Ref | |
| $50,000 to $99,999 | 0.73 | 0.65, 0.83 | <0.001 |
| $100,000 or more | 0.62 | 0.55, 0.71 | <0.001 |
| Prefer not to answer | 0.80 | 0.63, 1.00 | 0.005 |

IRR: Incidence Rate Ratio, CMIA: COVID-19 Mortality Impacted Areas

\*Higher PHQ-2 score (0–6) indicates an increased risk of major depression

\*\*White: non-Hispanic White; Non-White: Hispanic, Black, Asian, Native Hawaiian and Pacific Islander and American Indian and Alaska Native

## Discussion

Soon after Los Angeles County reached its peak monthly COVID-19 mortality rate in January 2021, residents in regions experiencing higher COVID-19 mortality rates reported a greater increase in the proportion of adults being at risk for major depression than those in areas with lower COVID-19 mortality rates. While regions with higher COVID-19 mortality rates were characterized by a larger proportion of Non-White, younger residents with higher COVID-19 positivity, these sociodemographic characteristics did not explain away regional differences in depression risk in our multivariable analysis. In addition, while there were pre-existing mental health disparities across the three CMIAs, mental health worsened from 2018 to 2021, particularly among Non-White residents in the high CMIA, which were the regions that had a larger increase in mortality in Non-White population.

There are several potential explanations for the higher burden of mental health in the high CMIA, especially for Non-White residents. First, the increased burden was consistent with the effects of the pandemic on mortality, whereas Non-White residents in high CMIA experienced the largest increase in mortality. This area contained the highest concentration of Hispanic and non-Hispanic Black residents of low socioeconomic status, and these residents experienced disproportionately high rates of COVID-19 infection and mortality [23]. Many Hispanic and non-Hispanic Black residents worked as essential workers in jobs that often required working in close quarters which increased the chance of infection [34]. Hispanic residents in

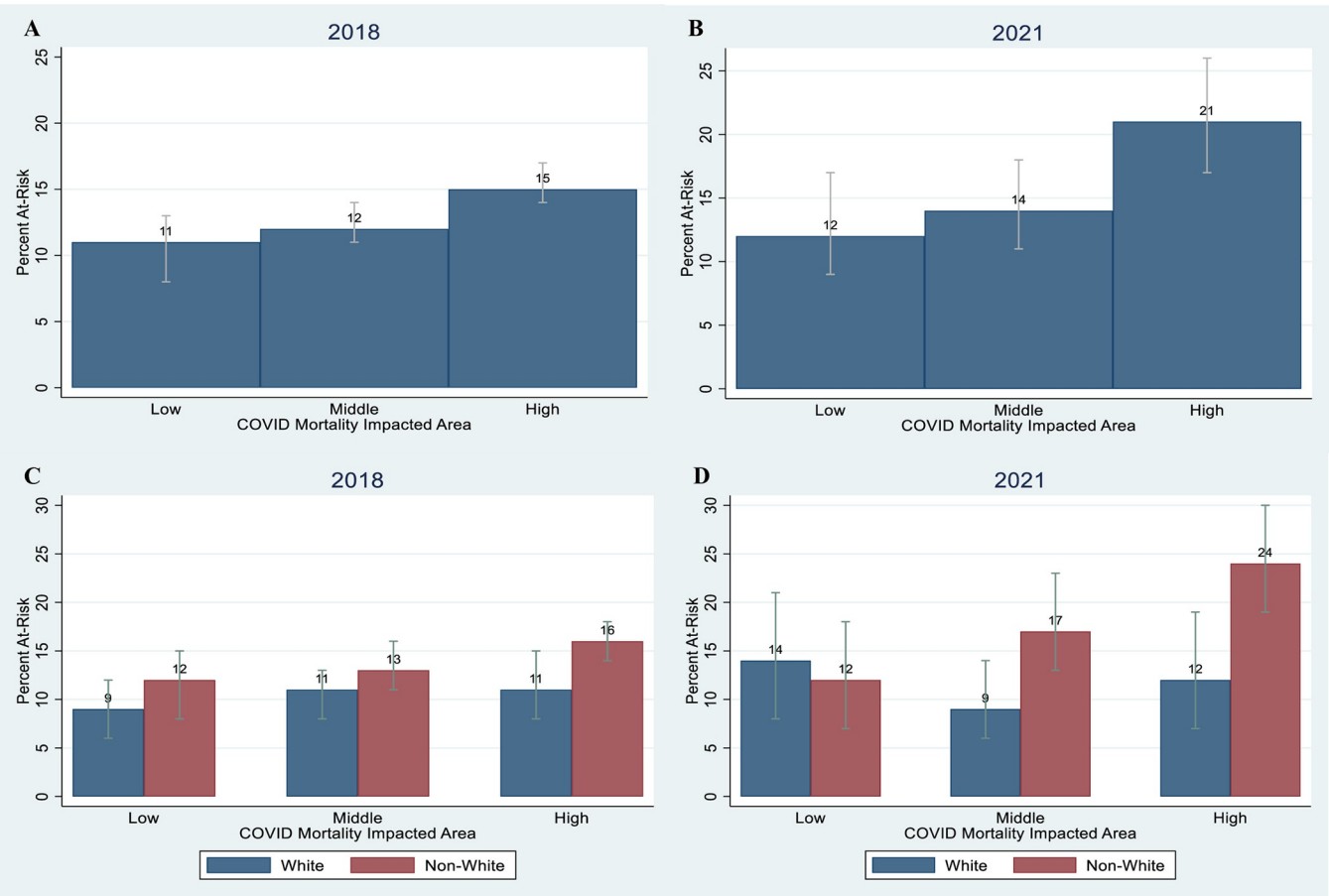

**Fig 1. At risk for major depression by CMIA and white vs. non-white residents\*, 2018 and 2021.** The percentage at risk reflects on the sample proportion with PHQ-2 sum score categorized as high (3–6). The grey bars show the confidence interval of the sample proportion. \*White: non-Hispanic White; Non-White: Hispanic, Black, Asian, Native Hawaiian and Pacific Islander and American Indian and Alaska Native. Fig 1A and 1B are overall percentages at each COVID-19 Mortality Impacted Areas (CMIA). Fig 1C and 1D are percentages at each CMIA by White vs Non-White Residents. Source 1: 2018 data from 2018 Los Angeles County Health Survey. Source 2: 2021 data from Los Angeles Pandemic Surveillance Cohort Study (current study).

the high CMIA also lived in multi-generational households where elder relatives might have suffered severe consequences if infected by a family member who was infected as an essential worker, leading to increased stress and anxiety [35]. Further, the worsening of mental health might also be due to an increase in personal stories of infection, hospitalization, and death among friends, family and neighbors, along with media coverage of the disproportionate impact of the pandemic on communities of color, which could have exacerbated mental health impacts on all residents [36, 37].

In addition, there was disproportionality in the ability of residents in the high CMIA to comply with COVID-19 mitigation strategies such as social distancing, quarantine and isolation, potentially due to their limited housing and financial resources [7, 24, 38]. Residents living in close quarters or multigenerational households in these areas likely faced greater difficulty in following public health mandates, potentially resulting in increased stress [3]. More research on how the mental health of residents of underserved communities was affected by the COVID-19 pandemic is warranted.

While Non-White residents in high CMIA experienced the largest deterioration in mental health, White residents in low CMIA demonstrated an unexpected increase in mental health

burden as well. This may be due to a greater comfort in reporting mental health problems among White residents in higher income areas due to lower stigma associated with these problems compared to what is experienced by racial/ethnic minority populations in our sample [39–41]. However, the finding on White residents in high CMIA should be interpreted with caution given the difference was not statistically significant.

This study has several limitations. First, the 2018 pre-pandemic mental health data are from a different survey population. The difference in sampling strategy and study population may result in two non-directly comparable samples. Second, due to limited sample size we were only able to evaluate differences in the impact of the pandemic for White versus Non-White residents. Asian residents, who had lower all-cause mortality rates than Black and White residents in Los Angeles County in general and had lower COVID-19 mortality rates than other racial/ethnic minority groups, were included in the Non-White category. This approach might miss substantial heterogeneity in the changes in mental health from 2018 to 2021, and by CMIA, among Non-White residents. We conducted a sensitivity analysis by removing Asian from the Non-White group to account for this effect, and the result remained the same. Third, the SPAs that we used to generate the CMIAs are large geographic areas with heterogeneity and diversity within each SPA itself, which could also impact the analysis. Fourth, survey nonresponse may have resulted in selection bias. To mitigate this potential problem, we weighted our analytic sample to represent the demographic characteristics in Los Angeles County.

## Conclusion

Living in an area with higher COVID-19 mortality rates may have been associated with worse mental health. These associations may vary by race/ethnicity, with Non-White residents reporting worse mental health outcomes in the high mortality area. The findings suggest that there may have been a worsening of mental health outcomes for some groups from 2018 to 2021. The findings highlight the need for greater public and mental health resources in areas with high COVID-19 mortality and among racial/ethnic minority groups.

## Author Contributions

**Conceptualization:** Chun Nok Lam, William Nicholas, Jennifer B. Unger, Neeraj Sood.

**Data curation:** Chun Nok Lam, Alex Ho.

**Formal analysis:** Chun Nok Lam, Alex Ho, Neeraj Sood.

**Funding acquisition:** Neeraj Sood.

**Investigation:** Chun Nok Lam, William Nicholas, Jennifer B. Unger, Neeraj Sood.

**Methodology:** Chun Nok Lam, William Nicholas, Jennifer B. Unger, Neeraj Sood.

**Supervision:** William Nicholas, Jennifer B. Unger, Neeraj Sood.

**Validation:** Chun Nok Lam.

**Visualization:** Chun Nok Lam, Neeraj Sood.

**Writing – original draft:** Chun Nok Lam, Ryan Lee, Daniel Soto, William Nicholas, Neeraj Sood.

**Writing – review & editing:** Chun Nok Lam, Ryan Lee, Daniel Soto, William Nicholas, Jennifer B. Unger, Neeraj Sood.

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
