## [Decision Letter · Decision Letter 0]

20 May 2024

The Differential Impacts of COVID-19 Mortality on Mental Health by Residential Geographic Regions: The Los Angeles Pandemic Surveillance Cohort Study

PONE-D-24-11827

Dear Dr. Lam,

We’re pleased to inform you that your manuscript has been judged scientifically suitable for publication and will be formally accepted for publication once it meets all outstanding technical requirements.

Kind regards,

Souparno Mitra, M.D.

Academic Editor

PLOS ONE

2. In this instance it seems there may be acceptable restrictions in place that prevent the public sharing of your minimal data. However, in line with our goal of ensuring long-term data availability to all interested researchers, PLOS’ Data Policy states that authors cannot be the sole named individuals responsible for ensuring data access (http://journals.plos.org/plosone/s/data-availability#loc-acceptable-data-sharing-methods).

Reviewers' comments:

Reviewer's Responses to Questions

**Comments to the Author**

1. Is the manuscript technically sound, and do the data support the conclusions?

Reviewer #1: Yes

Reviewer #2: Yes

2. Has the statistical analysis been performed appropriately and rigorously? 

Reviewer #1: Yes

Reviewer #2: Yes

3. Have the authors made all data underlying the findings in their manuscript fully available?

Reviewer #1: No

Reviewer #2: Yes

4. Is the manuscript presented in an intelligible fashion and written in standard English?

Reviewer #1: Yes

Reviewer #2: Yes

5. Review Comments to the Author

Reviewer #1: Overall, this is a clear, and well-written manuscript. The introduction is relevant. This is an interesting study and the data are informative. There are numerous strengths to this study. The study seems well conducted. The manuscript is structured. It is appropriate in length.

Reviewer #2: The article helped in understanding the specific mental health impacts among different ethnicities within the regions of most affected by COVID-19 mortality. While the pandemic has affected the entire population in general this article can identify the vulnerable communities. Some of the limitations appear to be studies limited to specific geographical areas and screening tools of PHQ-2 versus PHQ-9which can give a better understanding of mental health impacts.

6. PLOS authors have the option to publish the peer review history of their article (what does this mean?). If published, this will include your full peer review and any attached files.

Reviewer #1: No

Reviewer #2: **Yes: **Rajasekhar Kannali

---

## [Editor Report · Acceptance letter]

28 Jun 2024

PONE-D-24-11827 

PLOS ONE

Dear Dr. Lam, 

I'm pleased to inform you that your manuscript has been deemed suitable for publication in PLOS ONE. Congratulations! Your manuscript is now being handed over to our production team.

Kind regards, 

on behalf of

Dr. Souparno Mitra 

Academic Editor

PLOS ONE